# Inferring learning rules from animal decision-making

Zoe C. Ashwood[1,2,*]    Nicholas A. Roy[1,*]    Ji Hyun Bak[3,†]

The International Brain Laboratory    Jonathan W. Pillow[1,4]

[1]Princeton Neuroscience Institute, Princeton University
[2]Dept. of Computer Science, Princeton University
[3]Redwood Center for Theoretical Neuroscience, UC Berkeley
[4]Dept. of Psychology, Princeton University
{zashwood, nroy, pillow}@princeton.edu,
jihyun.bak@ucsf.edu, info@internationalbrainlab.org

## Abstract

How do animals learn? This remains an elusive question in neuroscience. Whereas reinforcement learning often focuses on the design of algorithms that enable artificial agents to efficiently learn new tasks, here we develop a modeling framework to directly infer the empirical learning rules that animals use to acquire new behaviors. Our method efficiently infers the trial-to-trial changes in an animal's policy, and decomposes those changes into a learning component and a noise component. Specifically, this allows us to: (i) compare different learning rules and objective functions that an animal may be using to update its policy; (ii) estimate distinct learning rates for different parameters of an animal's policy; (iii) identify variations in learning across cohorts of animals; and (iv) uncover trial-to-trial changes that are not captured by normative learning rules. After validating our framework on simulated choice data, we applied our model to data from rats and mice learning perceptual decision-making tasks. We found that certain learning rules were far more capable of explaining trial-to-trial changes in an animal's policy. Whereas the average contribution of the conventional REINFORCE learning rule to the policy update for mice learning the International Brain Laboratory's task was just 30%, we found that adding baseline parameters allowed the learning rule to explain 92% of the animals' policy updates under our model. Intriguingly, the best-fitting learning rates and baseline values indicate that an animal's policy update, at each trial, does not occur in the direction that maximizes expected reward. Understanding how an animal transitions from chance-level to high-accuracy performance when learning a new task not only provides neuroscientists with insight into their animals, but also provides concrete examples of biological learning algorithms to the machine learning community.

## 1   Introduction

Learning is a fundamental aspect of animal behavior, as it enables flexible adaptation to the time-varying reward structure of an environment. The ability of animals to learn new tasks also happens to be a fundamental component of neuroscience research: many experiments require training animals to perform a decision-making task designed to test specific theories of brain function. A deeper

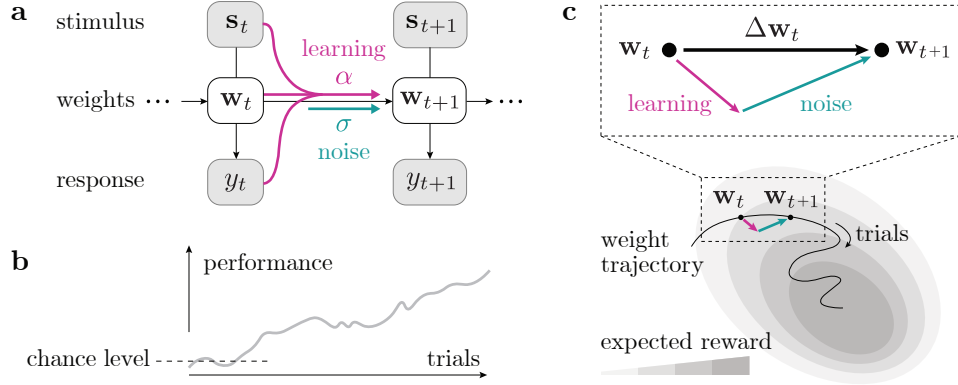

Figure 1: Model schematic. **(a)** We use a state-space representation with a set of time-varying weights $\mathbf{w}_t$, whose change is driven by a learning process as well as noise. **(b)** Animals usually improve their task performance with continued training, such that their expected reward gradually increases; however, the trial-to-trial change of behavior is not always in the reward-maximizing direction. **(c)** Considering the animal's learning trajectory in weight space, we model each step $\Delta\mathbf{w}_t$ as a sum of a learning component (ascending the expected reward landscape) and a random noise component.

understanding of how an animal learns—specifically, how it updates its policy in order to progress from chance-level to high-accuracy task performance—would provide both theoretical and practical benefits for understanding animal behavior, and would allow us to compare biological learning with the learning behavior of artificial agents [32].

Reinforcement learning typically focuses on the design of normative learning algorithms [7, 18, 30], which describe the optimal action selection policy for a given objective function [18]. These models have been successful in endowing artificial agents with the ability to efficiently learn complex tasks [16, 28, 33]; in the context of animal and human behavior, these models have successfully described and predicted various aspects of observed behavior [7, 18, 22, 29].

However, one of the greatest challenges for normative modeling lies in inferring an animal's internal model from behavior alone. Specifically, it is difficult to characterize an animal's objective function because animals often behave in ways that would not appear to increase their expected reward [2, 6, 12]. Modeling the intricacies of such behaviors often requires incorporating extensions into traditional reinforcement learning models, necessarily trading off the reward-maximizing properties of these models in order to better describe the data [9, 12, 17, 36]. Just as the rise of large public datasets enabled a revolution in machine learning [8, 13], the advent of high-throughput animal training opens the door to an exciting alternative approach to modeling learning. Leveraging new behavioral datasets containing millions of choices [11, 20], it may now be possible to infer *directly from data* the rules that govern animal learning.

In this paper, we present a flexible framework for inferring learning rules directly from an animal's decision-making behavior. Our method efficiently tracks the dynamic trial-to-trial changes in an animal's policy parameters over the course of training, and provides an interpretable decomposition of those changes into a learning component and an additive noise component (Fig. 1). Our method is formulated as a general framework that can model a variety of conventional learning rules, and offers a new way to compare different models of learning. We expect this approach to advance our understanding of learning from both computational and biological perspectives, offering valuable insights to neuroscientists and machine learning researchers alike.

## 2 Models and Methods

### 2.1 Model of decision-making

We present our model in the context of a generic two-alternative forced choice (2AFC) task, a common paradigm for studying decision-making behavior. On each trial, an animal is presented with a stimulus $\mathbf{s}_t$, and makes a choice $y_t \in \{0, 1\}$, referred to generally as a "leftward" or "rightward"

choice. An animal receives a reward ($r_t = 1$) if its response $y_t$ is correct, but receives no reward ($r_t = 0$) otherwise. The correct answer $\bar{y}_t$ for a given stimulus $\mathbf{s}_t$ depends on the rules of the specific task.

We assume that an animal's behavior on a trial $t$ is governed by an internal model, parametrized by a set of weights $\mathbf{w}_t$. These weights describe how an animal's choice depends on the task stimulus, $\mathbf{s}_t$, as well as any task-irrelevant aspects, such as a stimulus-independent bias toward one choice over the other. Specifically, we construct an input carrier vector $\mathbf{x}_t = [1, \mathbf{s}_t]$ that couples linearly with the weight vector $\mathbf{w}_t$, such that the probability of the animal going "rightward" on trial $t$ is

$$p(y_t = \text{rightward} \mid \mathbf{x}_t, \mathbf{w}_t) \equiv p_R(\mathbf{x}_t, \mathbf{w}_t) = \frac{\exp(\mathbf{x}_t \cdot \mathbf{w}_t)}{1 + \exp(\mathbf{x}_t \cdot \mathbf{w}_t)}. \tag{1}$$

The weight component that interacts with the constant "1" in the input vector captures the bias to choose to go "rightward", while the weights that couple with the task stimuli $\mathbf{s}_t$ represent the animal's stimulus sensitivities. This model is generally applicable to a wide array of tasks and can be readily extended to include other types of task-irrelevant covariates, such as the history dependence [4, 23, 24]; it is also a natural extension of the classic psychometric curve approach [34] to modeling choice behavior.

## 2.2 Model of trial-to-trial weight update

Because learning is inherently a time-varying process, we need a way to characterize how the weights evolve over time. We use a state-space representation for the weights $\mathbf{w} \in \mathbb{R}^K$ [1, 4, 19], in which we model the weight update $\Delta \mathbf{w}_t$ as a function of the input $\mathbf{x}_t$, the output $y_t$, as well as the current state $\mathbf{w}_t$ (Fig. 1a).

We assume that the weight update $\Delta \mathbf{w}_t$ can be decomposed into a deterministic learning component, $\mathbf{v}_t$, and an independent Gaussian innovation noise, $\boldsymbol{\eta}_t$:

$$\Delta \mathbf{w}_t \equiv (\mathbf{w}_{t+1} - \mathbf{w}_t) = \mathbf{v}_t + \boldsymbol{\eta}_t, \qquad \boldsymbol{\eta}_t \sim \mathcal{N}(0, \text{diag}(\sigma_1^2, \ldots, \sigma_K^2)). \tag{2}$$

The variance of the noise component for the $k$-th weight is captured by $\sigma_k^2$, the *volatility* hyperparameters. The deterministic component is modeled with a specific learning rule, scaled by a non-negative learning rate; for now we simply use $\mathbf{v}_t$ as a placeholder variable to write

$$\mathbf{v}_t = \text{diag}(\alpha_1, \ldots, \alpha_K)\, \hat{\mathbf{v}}_t, \qquad \hat{\mathbf{v}}_t = \texttt{LearningRule}(\mathbf{w}_t, \{\mathbf{x}_t, y_t\}, r_t). \tag{3}$$

Element-wise, this is equivalent to $[\mathbf{v}_t]_k = \alpha_k [\hat{\mathbf{v}}_t]_k$ for the $k$-th component of the weight vector. In general, we can treat each *learning rate* $\alpha_k$ as a separate hyperparameter in the model.

We note that our method is an extension of our earlier work [23, 24], which presented a purely descriptive model that used only the noise term, $\boldsymbol{\eta}_t$, to capture trial-to-trial changes in the weight trajectory.

## 2.3 Learning models

Our framework allows flexible exploration of different learning models, as long as the predicted weight update due to learning, $\mathbf{v}_t$, can be computed from the current weights $\mathbf{w}_t$ and any past experience (such as previous choices $y_{1:t}$ or rewards $r_{1:t}$). In particular, policy-gradient learning [7, 15, 30, 31] fits readily into our framework.

In this paper, we explore the family of REINFORCE [35] learning rules, a set of well-established policy update rules that seek to maximize expected reward by sampling the policy gradient at each trial. In its simplest form, the REINFORCE update for the $k$-th weight is given by

$$[\mathbf{v}_t]_k = \alpha_k \cdot r_{a_t, \bar{y}_t} \cdot \epsilon_{a_t}(1 - p_{a_t})[\mathbf{x}_t]_k, \qquad \epsilon_R = +1, \ \epsilon_L = -1, \tag{4}$$

where $a_t$ is the animal's choice at trial $t$, $\bar{y}_t$ is the correct answer for this trial and $p_{a_t}$ is the probability, as obtained from the policy, that the animal selected action $a_t$ at trial $t$ ($p_R(\mathbf{x}_t, \mathbf{w}_t)$ from Eq. 1 in the case that the animal selected to go rightward, and $(1 - p_R(\mathbf{x}_t, \mathbf{w}_t))$ in the case that the animal went to the left at trial $t$). The reward is $r_{a, \bar{y}} = 1$ when $a = \bar{y}$ (correct choice), and 0 otherwise (incorrect). The sign $\epsilon_a$ is simply a mathematical consequence of how we modeled the choice probability (Eq. 1). In what follows, we will refer to the model defined by the combination of Eq. 1, Eq. 2 and the

REINFORCE learning rule of Eq. 4 as $\text{RF}_K$. In the case that there is a single $\alpha$ parameter shared across the $K$ dimensions in Eq. 4, we will refer to this model as $\text{RF}_1$.

We also consider a version of REINFORCE with a constant but weight-specific baseline, $\{\beta_k\}$ [30, 35]:

$$[\mathbf{v}_t]_k = \alpha_k \cdot (r_{a_t, \bar{y}_t} - \beta_k) \cdot \epsilon_{a_t}(1 - p_{a_t})[\mathbf{x}_t]_k. \tag{5}$$

Note that the baseline adjusts the *effective reward* in the update equation, in the sense that the reward term $r$ in Eq. 4 is now replaced by $(r - \beta_k)$. Each baseline $\beta_k$ is an additional hyperparameter in the model, to be inferred data. We will refer to the model defined by this learning rule (and Eq. 1, Eq. 2) as $\text{RF}_\beta$. See the Supplementary Material (SM) for a case-by-case evaluation of both the REINFORCE and REINFORCE with baseline learning rules.

## 2.4 Inference of the weight trajectory

We use hierarchical Bayesian inference to estimate both the time-varying weight trajectory and the best set of hyperparameters from choice data. The inference procedure consists of two loops, for (i) weight estimate at fixed hyperparameters, and (ii) hyperparameter optimization.

For the *inner* loop, the goal is to determine the combined weight trajectory that maximizes the posterior distribution, at fixed hyperparameters $\phi \equiv \{\sigma_1, \ldots, \sigma_K, \alpha_1, \ldots, \alpha_K, \beta_1 \ldots, \beta_K\}$. With $T$ trials in the dataset, and $K$ weights, we optimize the entire weight trajectory at once by representing it as a $KT$-dimensional vector $\mathbf{w}$. The posterior is constructed from the likelihood function and the prior distribution. The model of choice probability (Eq. 1) specifies the trial-specific likelihood function $p(y_t|\mathbf{x}_t, \mathbf{w}_t)$. The model of trial-to-trial weight updates (Eqs. 2-3) corresponds to a Gaussian prior on the weight trajectory, $\mathbf{w} \sim \mathcal{N}(\mathbf{u}, C)$, with mean $\mathbf{u} = D^{-1}\mathbf{v}$ and covariance $C^{-1} = D^\top \Sigma^{-1} D$; here $D$ is the difference matrix constructed as $K$ copies of a $T \times T$ matrix stacked block-diagonally, where each $T \times T$ block has $+1$ along the main diagonal and $-1$ along the lower off-diagonal; $\Sigma$ is a diagonal matrix of $\sigma$'s (see SM for full details) [4]. Note that the prior mean $\mathbf{v}$ depends on the hyperparameters $\{\alpha_k\}$ and $\{\beta_k\}$, and the covariance matrix $C$ depends on $\{\sigma_k\}$. Given data $\mathcal{D} = \{\mathbf{x}_t, y_t\}_{t=1,\ldots,T}$, the log-posterior for $\mathbf{w}$ is

$$\log p(\mathbf{w}|\mathcal{D}; \phi) = \tfrac{1}{2}\big(\log |C|^{-1} - (\mathbf{w} - \mathbf{u})^\top C^{-1}(\mathbf{w} - \mathbf{u})\big) + \sum_{t=1}^{T} \log p(y_t|\mathbf{x}_t, \mathbf{w}_t) + const. \tag{6}$$

For the *outer* loop, we perform numerical optimization so as to obtain the set of hyperparameters, $\phi = \{\{\sigma_k\}, \{\alpha_k\}, \{\beta_k\}\}$, that maximize the approximate marginal likelihood, or *evidence*, defined as $p(\mathcal{D}|\phi) = \int d\mathbf{w}\, p(\mathcal{D}|\mathbf{w})p(\mathbf{w}|\phi) = p(\mathcal{D}|\mathbf{w})p(\mathbf{w}|\phi)/p(\mathbf{w}|\mathcal{D}; \phi) \approx \frac{p(\mathcal{D}|\mathbf{w})p(\mathbf{w}|\phi)}{\mathcal{N}(\mathbf{w}|\mathbf{w}_{\text{MAP}}, -H^{-1})}$ [5, 25]. We use Laplace approximation to approximate the posterior distribution over the weights, $p(\mathbf{w}|\mathcal{D}; \phi) \approx \mathcal{N}(\mathbf{w}|\mathbf{w}_{\text{MAP}}, -H^{-1})$, where $H$ is the Hessian at the maximum a posteriori estimate of the weights, $\mathbf{w}_{\text{MAP}}$. We usually fit multiple models with different hyperparameter initializations to locate the global optimum of model evidence. See SM for full details of the inference procedure.

# 3 Results

Although our method can be applied to a wide variety of tasks, for clarity, we will mostly focus on a specific 2AFC task: the International Brain Laboratory's (IBL) decision-making task based on visual detection [11]. See Fig. 2a and the caption for a more in-depth description of the task.

## 3.1 We can infer learning rates from simulated data

We first demonstrate our method with simulated data that resembles a mouse's choices as it learns the IBL task. We generated two time-varying weight trajectories for the mouse's bias and sensitivity (Fig. 2b), according to the trial-to-trial weight update model (Eq. 2) with the standard REINFORCE learning rule (Eq. 4). Four hyperparameters were used to generate the weight trajectories, with a learning rate $\alpha$ and a noise strength $\sigma$ for each weight (Fig. 2c); the values were chosen to emulate the properties of the real data. We used the probabilistic decision-making model (Eq. 1) to generate a stimulus-response pair for each trial based on the weights, and only used this simulated behavioral data to infer the weight trajectories and the hyperparameters.

We confirm that our method accurately recovers the weight trajectories (Fig. 2b), as well as the values of the underlying hyperparameters (Fig. 2c). In particular, it is able to separately infer different values

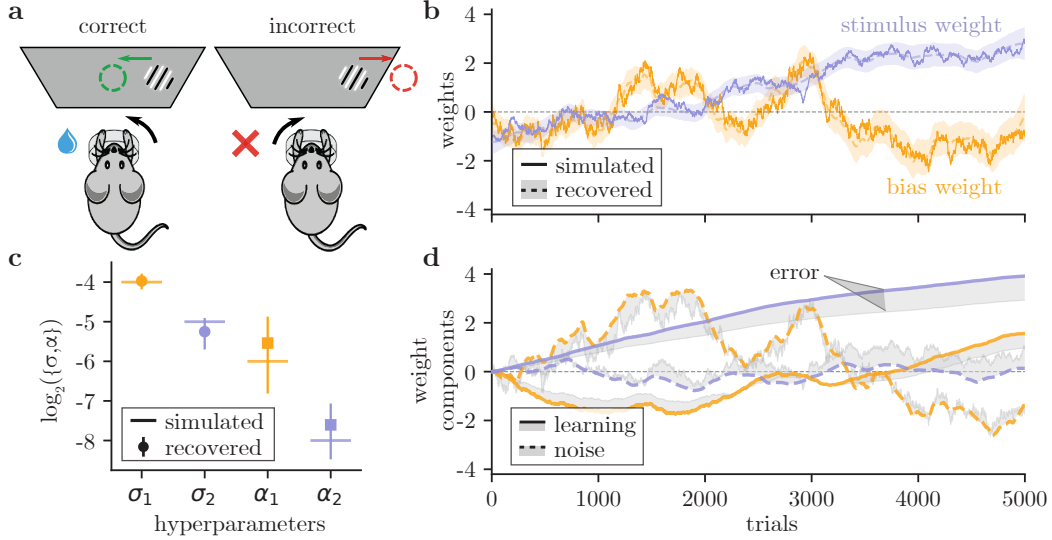

Figure 2: Validation on simulated data. **(a)** The IBL task [11]: on each trial, a sinusoidal grating (with contrast values between 0 and 100%) appears on either the left or right side of a screen. Mice must report the side of the grating by turning a wheel (left or right) in order to receive a water reward. **(b)** We simulate a bias weight and stimulus weight (solid lines) which evolve according to our model using the REINFORCE rule, then generate choice data. From the choice data, we successfully recover the weights (dashed lines) with a 95% credible interval (shading). **(c)** We also successfully recover the underlying hyperparameters from the simulated data (error bars are $\pm 1$ posterior SD).**(d)** We decompose each recovered weight into a learning component (solid lines) and a noise component (dashed lines). Shading shows the cumulative error between the true and recovered components.

for the learning rates ($\alpha_1$ and $\alpha_2$) for the two different weights. In the SM, we show additional recovery analyses, including the recovery of $\text{RF}_\beta$ weights and hyperparameters from simulated data.

### 3.2 We can decompose weight updates into learning and noise

Moreover, our method can explicitly decompose each trial-to-trial weight change $\Delta\mathbf{w}_t$ into a learning component $\mathbf{v}_t$ and a noise component $\boldsymbol{\eta}_t$ (Eq. 2). Fig. 2d demonstrates that our method has accurately de-mixed the components separately, as shown by the narrow shaded gap between the true and retrieved curves. The decomposition provides a useful tool for interpreting the behavioral data. In particular, we can quantify how much of the trial-to-trial weight update is along the dimension of learning: we calculate the projected square magnitude of the update $|\Delta\mathbf{w}_{\hat{\mathbf{v}},t}|^2 = (\Delta\mathbf{w}_t \cdot \hat{\mathbf{v}}_t)^2/|\hat{\mathbf{v}}_t|^2$ that is parallel to $\hat{\mathbf{v}}_t$, and its fraction out of the net weight update $|\Delta\mathbf{w}_t|^2$. In the simulated data (Fig. 2), for example, the average contribution from the learning dimension is $\langle|\Delta\mathbf{w}_{\hat{\mathbf{v}},t}|^2/|\Delta\mathbf{w}_t|^2\rangle_{\text{all trials}} = 0.53$. This is a better approach than a naive comparison between the inferred values of the learning rate $\alpha$ and the noise strength $\sigma$, which can be difficult to interpret.

In the SM, we explore the effect of model mismatch on both hyperparameter and weight decomposition recovery. There, we show that a successful decomposition of learning and noise components depends crucially on the correct choice of the learning model. In this case, we know that the learning model for the inference is *correct*, because it assumes the same model (REINFORCE) as the one used for generating the "true" weight trajectory. In the case of actual data, the true learning model is not known; however, as we also show in the SM, the Akaike Information Criterion, can help us successfully identify the underlying learning model (see SM).

### 3.3 Animals learn different weights at different rates

Now we apply our method to model the choice behavior of 13 mice (78,000 trials; 6,000 trials per mouse) learning the IBL task as they transition from chance-level to greater than 70% accuracy (see SM for learning curves for these animals). Taking one mouse as an example, we plot the weight

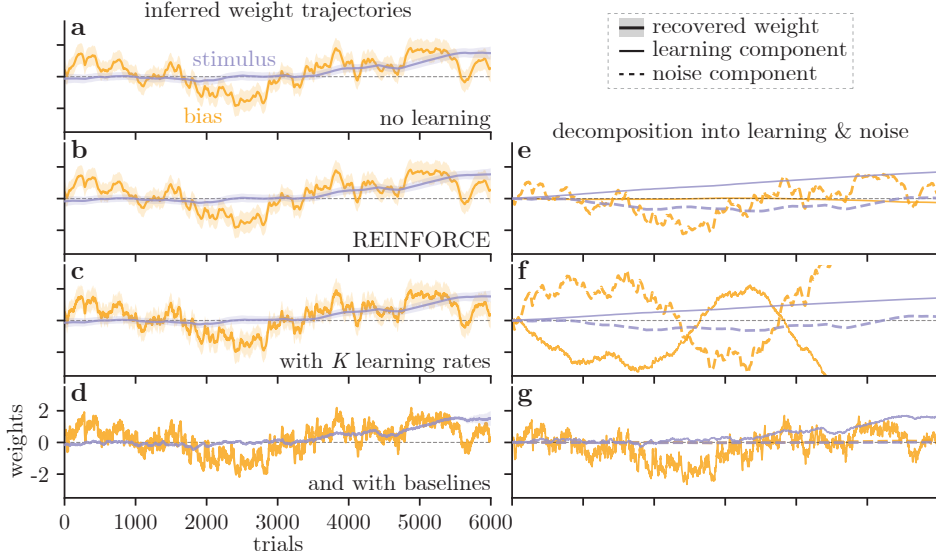

Figure 3: Result from an example IBL mouse. **(a-d)** Inferred trial-to-trial weight trajectories for the choice bias (yellow) and contrast sensitivity (purple), recovered under different learning models: **(a)** $\text{RF}_0$, No learning model, with only a noise component to *track* the changes in behavior with the noise component. This mouse's bias fluctuates between leftward and rightward choices (negative and positive bias weight), whereas its decision-making is increasingly influenced by the task stimuli (gradually increasing stimulus weight). **(b)** $\text{RF}_1$, REINFORCE with a single learning rate for all weights. **(c)** $\text{RF}_K$, REINFORCE with a separate learning rate for each of the two weights. **(d)** $\text{RF}_\beta$, REINFORCE with baselines, where the baseline is also inferred separately for each weight. **(e-g)** The decomposition of trial-to-trial weight updates into learning and noise components, for the model shown in the same row. The noise component is shown with the dashed line, while the learning component is given by the solid line.

trajectories inferred under different learning models (Fig. 3). Specifically, we compared a model without learning ($\text{RF}_0$; Fig. 3a); a model with REINFORCE learning and a single learning rate $\alpha$ for all weights ($\text{RF}_1$; Fig. 3b); and a model with REINFORCE learning and separate learning rates for all $K = 2$ weights ($\text{RF}_K$; Fig. 3c). We also performed the same analysis for the 13 mouse cohort (see SM for weight decompositions for whole cohort). We measure the model fit in terms of the Akaike Information Criterion (AIC), or the negative log-likelihood penalized by the number of parameters. For the majority of our cohort, the model fit is considerably better (the AIC is smaller) for $\text{RF}_K$ compared to both $\text{RF}_0$ and $\text{RF}_1$ (Fig. 4d).

Allowing per-weight learning rates has interesting implications. When the learning rate is uniform ($\text{RF}_1$), the prescribed weight update is in the direction of the gradient of the expected reward, and therefore follows the path of steepest ascent in the reward landscape. On the other hand, if the learning rate is different for each direction ($\text{RF}_K$), the weight update is not necessarily along the direction of the gradient. We show simulated trajectories for a mouse using the $\text{RF}_1$ and the $\text{RF}_K$ learning rules to update its policy ($\sigma = 0$ in this case) in Fig. 5 (lines (b) and (c), respectively). That the AIC for the $\text{RF}_K$ model is so much better than that for the $\text{RF}_1$ model, and that the retrieved weight trajectory for the animal (Fig. 5a) looks so different to the $\text{RF}_1$ trajectory, suggests that the learning behavior of the mouse cannot be explained as a simple gradient ascent on the expected reward landscape.

## 3.4 Animals shift their effective reward when learning

Whereas the net trajectories from $\text{RF}_1$ and $\text{RF}_K$ look very similar (Fig. 3b-c), they give different results for the learning-noise decomposition (Fig. 3e-f), especially if we focus on the bias (yellow). For $\text{RF}_1$, it is the noise that captures most of the changes in the bias, while the contribution of the learning component is negligible (Fig. 3e). For $\text{RF}_K$, the learning component for the bias is more dynamic, but its cumulative effect deviates more significantly from the inferred trajectory (Fig. 3f).

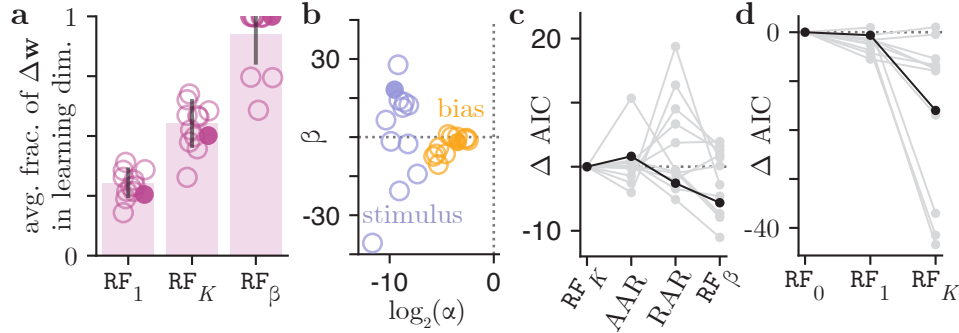

Figure 4: Population analysis for 13 IBL mice. **(a)** The average fraction of the trial-to-trial weight updates along the learning direction, as prescribed by three learning models $RF_1$, $RF_K$, and $RF_\beta$. Each open circle represents a mouse; the example mouse from Fig. 3 is marked by a filled circle. The solid bars indicate the mean fraction across the animal cohort. Whereas the mean fraction of animals' weight updates due to learning is just 0.30 for the $RF_1$ model, it is 0.92 for the $RF_\beta$ model. **(b)** The inferred learning rates and baselines, for the contrast and bias weights, from each mouse using the $RF_\beta$ model. **(c)** Model comparison across learning rules within RF family, and beyond it (see Sec. 3.5 for a description of AAR and RAR learning rules), in terms of the difference in their Akaike Information Criterion (AIC) relative to the REINFORCE model ($RF_K$). Each line is a mouse, and our example mouse is marked in black. **(d)** Model comparison within the family of REINFORCE models, with different numbers of varied learning rates. One outlier mouse was excluded from this figure for visibility (the AIC decreased by a massive 126.5 for the $RF_K$ model relative to the $RF_0$ model). Our example mouse is marked in black.

The consistently larger slope of the cumulative learning, compared to the net weight trajectory, suggests that our learning model for the trial-to-trial weight change is missing an additive offset.

To introduce additive offsets to the learning component, we considered another learning model, RE-INFORCE with baseline ($RF_\beta$; Eq. 5), with separate learning rates and baselines for different weights. We used the $RF_\beta$ model to infer the weight trajectories (Fig. 3d), and plotted their decompositions into learning and noise components (Fig. 3g). Interestingly, the trial-to-trial weight changes are almost entirely captured by the learning components (Fig. 3g), which was consistent across the entire cohort of 13 mice (see SM).

From the inferred hyperparameters for the $RF_\beta$ model (Fig. 4b), we can make several observations. Firstly, for all animals in the cohort, the learning rate for the bias is larger than the learning rate for the weight on the contrast. This is consistent with the finding that allowing per-weight learning rates, as in the $RF_K$ model, leads to vast improvements in AIC compared to $RF_1$ models with a single learning rate. The larger learning rate associated with the bias indicates that mice adjust their bias on a faster timescale than they adjust the weight they place on the contrast (which can also be readily observed in the retrieved weight trajectories of Fig. 3a-d). Understanding why animals' choice biases fluctuate on such short timescales will be an interesting question to explore in future work.

Secondly, we look to values of the retrieved baseline parameters in Fig. 4b in order to postulate as to why the REINFORCE with baseline model, as opposed to any of the other learning models we consider, is capable of explaining the trial-to-trial weight updates used by real animals. We notice that the baseline values for all animals for both the stimulus and bias are non-zero. Recall that with $RF_\beta$, as given by Eq. 5, the effective reward for a correct trial is $(1 - \beta_k)$, and $\beta_k$ for an incorrect trial. Thus, compared to the $RF_1$ and $RF_K$ models, the $RF_\beta$ model results in non-zero weight updates for error trials. Furthermore, that the baseline values are not equal to 0.5 (since $r_{a_t, \bar{y}_t} \in \{0, 1\}$) indicates that error and correct trials result in updates of different sizes. Finally, the fact that the retrieved baseline values are often negative or, when they are positive, are greater than 1 allows for the sign of the weight update given in Eq. 5 to change for either correct trials (in the case that the baseline is larger than 1) or error trials (in the case that the baseline is negative). In this way, the $RF_\beta$ model may be better equipped to handle seemingly 'suboptimal' weight updates that do not seem to maximize expected reward. However, fully understanding why the $RF_\beta$ learning rule, as opposed to the $RF_K$

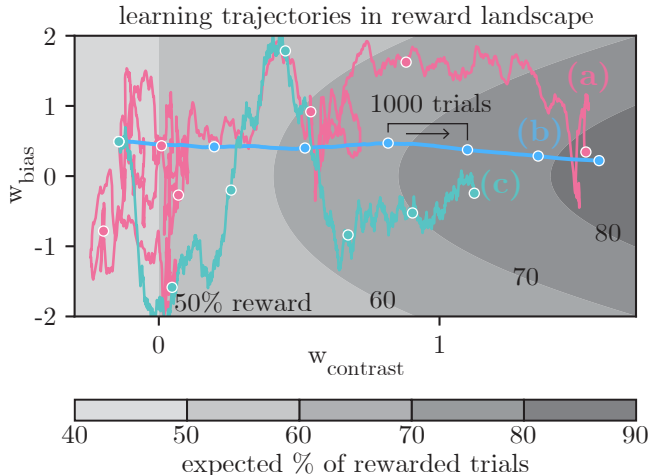

Figure 5: Weight trajectories plotted on the expected reward landscape for the IBL task. When the animal increases $w_{\text{contrast}}$ and simultaneously decreases $w_{\text{bias}}$ to zero, this results in a higher expected reward. **(a)** The recovered full trajectory for an example IBL mouse over the course of 6000 trials for the $\text{RF}_K$ model (this is the same trajectory that is shown in Figure 3c). We compare the animal's trajectory with deterministic trajectories generated (without noise) from the **(b)** $\text{RF}_1$ and **(c)** $\text{RF}_K$ learning rules when the learning rates are fixed to those inferred from data.

and $\text{RF}_1$ learning rules, explains the trial-to-trial weight changes of real animals will require further analyses and will be explored in future work.

### 3.5 Model comparison for different learning models

So far, our approach in this work was to only introduce new parameters if they improved the model (either in terms of the fit, or the interpretability); the lower AIC values obtained by allowing separate learning rates ($\text{RF}_K$), or adding baselines ($\text{RF}_\beta$), justified our modeling choices (Fig. 4c-d).

But the generality of model comparison supported by our method is not limited to the inclusion of additional parameters; any learning rule detailing trial-to-trial weight updates can be considered and plugged into Eq. 3. In addition to $\text{RF}_1$, $\text{RF}_K$ and $\text{RF}_\beta$, we compared two other learning rules that are closely related to REINFORCE, but make different assumptions. Specifically, we considered the *action-averaged* REINFORCE (AAR), $[\Delta\mathbf{w}]_k = \alpha_k \cdot p_{\bar{y}} \cdot \epsilon_{\bar{y}}(1 - p_{\bar{y}})[\mathbf{x}]_k$, where effective reward is averaged over the choice probability; and the *reward-agnostic* REINFORCE (RAR), $[\Delta\mathbf{w}]_k = \alpha_k \cdot \epsilon_{\bar{y}}(1 - p_{\bar{y}})[\mathbf{x}]_k$, whose effective reward is a constant 1 (see SM for more rationales). We find that $\text{RF}_\beta$ provided better fits to more animals, including our example animal (Fig. 4), although there was considerable variation in the preferred learning rule across animals. It will be an interesting future work to compare a broader variety of learning rules.

### 3.6 Application to other datasets

Our method offers a general method that can be applied to analyze choice behavior in many perceptual decision-making experiments. Here we analyze data from a different animal species (rat) learning a different task [2], where the stimulus is a delayed pair of two auditory tones (see Fig. 6 and caption for task details). Now we need three weights (bias and two tone sensitivities) to describe the decision-making behavior. We first recovered the three weight trajectories using a no-learning model (Fig. 6a), and performed similar analyses as before, using the three learning models $\text{RF}_1$, $\text{RF}_K$ and $\text{RF}_\beta$. As we show in the SM, the $\text{RF}_1$ and $\text{RF}_K$ models are once again incapable of explaining the trial-to-trial weight changes in the animals' policies; however, as shown in Fig. 6d, the $\text{RF}_\beta$ model is equipped to explain the majority of the trial-to-trial weight updates. See SM for the full set of results for this dataset.

## 4 Discussion

In this work, we develop a novel framework for extracting the learning rules underpinning animal behavior as mice and rats learn to perform perceptual decision-making tasks. Our method can accurately infer the time-varying weights governing an animal's policy, along with an animal's learning rates for different weights, hyperparameters governing the noise in different weights, as well as any reward baselines that the animal may be using in order to update its policy. We validated our method on simulated data, and applied it to a cohort of 13 mice learning to perform a sensory

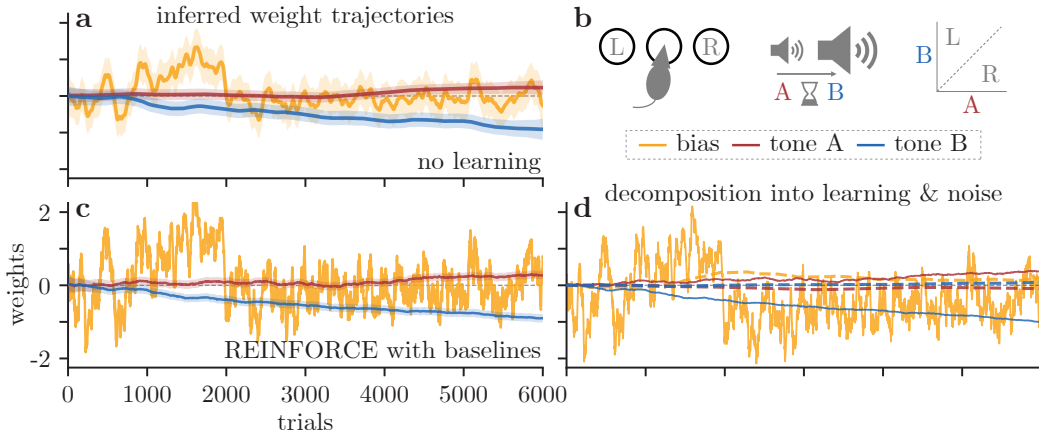

Figure 6: Results from a rat auditory discrimination task [2]. **(a)** We track an example rat's choice bias (yellow) and the sensitivity to two stimuli (red, blue) while training on the task described in (b). **(b)** In this task, a rat hears two tones of different amplitudes (tones A and B) separated by a delay. If tone A is quieter than B, the rat must nose-poke into the left port for reward, and vice-versa if tone A is louder than B. **(c)** We now use the $\mathrm{RF}_\beta$ model to predict how our rat updates its behavior. **(d)** The weights from (c) are decomposed into learning (solid) and noise (dashed) components, as in Fig. 3g.

decision-making task [11], as well as 19 rats learning a parametric working memory task [2]. In addition to the inferred learning rule and time-varying weights themselves, our method provides interpretable decompositions of the weight trajectories into learning and noise components. Whereas learning models are typically compared via generic model comparison metrics like the Akaike Information Criterion, comparing learning rules by their ability to explain the underlying trial-to-trial changes in an animal's policy provides an intuitive alternative.

Based on applications to two datasets, we were able to make several novel observations about animal learning, two of which we feel are particularly interesting. First, our best-fit model shows that different components of an animal's policy, controlled by different weights in our model, are updated with different learning rates. This implies that animal learning does not necessarily follow the gradient of the expected reward landscape at every trial. Although weight-specific learning rates $\alpha_k$ were formally proposed in [35], our work provides empirical evidence for weight-specific learning in real animals. Second, we found that the REINFORCE with baseline ($\mathrm{RF}_\beta$) model, with additive offsets to the effective reward, does particularly well in capturing the trial-to-trial weight changes along the dimension of prescribed learning. This finding was upheld in a second dataset involving rats learning a task with a two-dimensional stimulus space. Given that the prescribed learning rule affects only a 1-D subspace of the model weights, the success of $\mathrm{RF}_\beta$ to capture fluctuations in this higher-dimensional weight space is even more remarkable. Understanding of the role of baselines (effective reward offsets), as well as an investigation of whether they vary over time (as in, for example, the Actor-Critic framework [30]) present promising directions for future work.

We briefly discuss several limitations of our method. Our model of learning does not incorporate sensory uncertainty [14, 21], history dependence [2, 6, 10] or state dependence [3], all of which are known to affect decision-making behavior. Our model can, however, be readily extended to allow more flexible descriptions of learning rates and baselines, for example by introducing session-by-session changes for the hyperparameters [24]. Furthermore, an exciting future direction will be to compare our model of choice behavior with value function based models, such as variants of Rescorla-Wagner [22], or other dynamic models such as those considered in [26, 27]. While [15] provides support for the view that humans use policy-gradient methods instead of value prediction, Temporal Difference (TD) methods are more typically used to model choice behavior in the computational cognitive science community, and a comparison of our model with some standard TD models would help contextualize our work. Despite these limitations, we believe that our method can be readily applied to study different tasks, animal species, and learning models; and that it can be used to provide insights into empirical features of animal learning. We believe our approach will provide a powerful framework for the data-driven investigation of animal learning behavior.

## Broader Impact

Our work seeks to describe and predict the choice behavior of rodents in the context of decision-making experiments. We hope that neuroscientists and psychologists use our framework to better understand learning within their own experiments, and we have publicly released our code so as to enable this (https://github.com/pillowlab/psytrack_learning). Additionally, our work leverages data from two new publicly available datasets [2, 11], acting as an example of the value of open-science practices.

## Acknowledgments and Disclosure of Funding

We thank Peter Dayan, Yotam Sagiv and Sam Zorowitz for helpful comments and discussion at the beginning of this project. We thank Nathaniel Daw, Alejandro Pan-Vazquez, Yoel Sanchez Araujo and Ilana Witten for useful comments and discussion as this project was being completed. Finally, we thank the anonymous NeurIPS reviewers for their insightful comments and feedback.

JWP was supported by grants from the Simons Collaboration on the Global Brain (SCGB AWD543027), the NIH BRAIN initiative (NS104899 and R01EB026946), and a U19 NIH-NINDS BRAIN Initiative Award (5U19NS104648).

## Footnotes

†Current address: University of California, San Francisco

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
