[Supplementary Material]

# Supplementary material

## Contents

## A    Case-by-case evaluation of learning rules (extension of Section 2.3)

In Table S1, we provide a case-by-case evaluation of the REINFORCE [7], REINFORCE with baseline, Action-Averaged REINFORCE (AAR), and Reward-Agnostic REINFORCE (RAR) learning rules for every action taken $a_t$ and every correct response $\bar{y}_t$. Here, $p_R \equiv p_R(\mathbf{x}_t, \mathbf{w}_t) = \frac{\exp(\mathbf{x}_t \cdot \mathbf{w}_t)}{1+\exp(\mathbf{x}_t \cdot \mathbf{w}_t)}$ from Eq. 1, and all of the other notation is the same as in text.

Table S1: Update to the $k$-th weight component according to different learning rules, after each pair of a chosen action, $a_t$ and the correct action, $\bar{y}_t$. The trial index $t$ is omitted from the values in the table. Sub-tables represent four learning rules, (a) REINFORCE, (b) REINFORCE with baseline, (c) action-agnostic REINFORCE, and (d) reward-agnostic REINFORCE, the first two of which correspond to equations in text.

### a. REINFORCE (Eq. 4)

|  | $a_t = L$ | $a_t = R$ |
|---|---|---|
| $\bar{y}_t = L$ | $-\alpha_k(p_R)[\mathbf{x}]_k$ | $0$ |
| $\bar{y}_t = R$ | $0$ | $\alpha_k(1-p_R)[\mathbf{x}]_k$ |

### b. REINFORCE with baseline (Eq. 5)

|  | $a_t = L$ | $a_t = R$ |
|---|---|---|
| $\bar{y}_t = L$ | $-\alpha_k(1-\beta_k)p_R[\mathbf{x}]_k$ | $-\alpha_k\beta_k(1-p_R)[\mathbf{x}]_k$ |
| $\bar{y}_t = R$ | $\alpha_k\beta_k p_R[\mathbf{x}]_k$ | $\alpha_k(1-\beta_k)(1-p_R)[\mathbf{x}]_k$ |

### c. AAR

|  | $a_t = L$ | $a_t = R$ |
|---|---|---|
| $\bar{y}_t = L$ | $-\alpha_k p_R(1-p_R)[\mathbf{x}]_k$ | |
| $\bar{y}_t = R$ | $\alpha_k p_R(1-p_R)[\mathbf{x}]_k$ | |

### d. RAR

|  | $a_t = L$ | $a_t = R$ |
|---|---|---|
| $\bar{y}_t = L$ | $-\alpha_k(p_R)[\mathbf{x}]_k$ | |
| $\bar{y}_t = R$ | $\alpha_k(1-p_R)[\mathbf{x}]_k$ | |

## B   Inference of weight trajectory (extension of Section 2.4)

Here, we elaborate on our inference procedure for weights $\mathbf{w}$ and hyperparameters $\phi$. Our weights on a trial $t$ are the $K$-element vector $\mathbf{w}_t$ corresponding to the $K$-elements in the input vector on that trial, $\mathbf{x}_t$. We arrange all of the $\{\mathbf{w}_t\}$ into a single vector by concatenating the $k^{\text{th}}$ element from each $\mathbf{w}_t$ into a vector of length $T$, then concatenating those $K$ vectors into a single vector of length $KT$. That is, $\mathbf{w} = [w_{k=1,t=1}, \ldots, w_{1,T}, w_{2,1}, \ldots, w_{K,T}]$.

The prior distribution on our weights $\mathbf{w}$ is given according to $\mathcal{N}(\mathbf{u}, C)$ with mean $\mathbf{u} = D^{-1}\mathbf{v}$ and covariance $C^{-1} = D^{\top}\Sigma^{-1}D$. Here $D$ is a difference matrix, constructed as $K$ copies of a $T \times T$ matrix stacked block diagonally, where each $T \times T$ block has $+1$ along the main diagonal and $-1$ along the lower off-diagonal. Practically, $[\Delta \mathbf{w}]_t = [D\mathbf{w}]_t$, while $D^{-1}$ is effectively taking a cumulative sum. $\Sigma$ is a $KT \times KT$ diagonal matrix where the diagonal vector is assembled by replicating each of the $\sigma_k$ hyperparameters $T$-times, $\text{diag}(\Sigma) = [\sigma_1, \ldots, \sigma_1, \sigma_2, \ldots, \sigma_K]$. Using Eq. 1, we define the likelihood of the model as $L = \prod_{t=1}^{N} p(y_t|\mathbf{x}_t, \mathbf{w}_t)$.

Using log-prior and log-likelihood, we can numerically optimize the log-posterior w.r.t. $\mathbf{w}$ for a fixed set of hyperparameters $\phi$ (using the BFGS algorithm [5]). Defining our *maximum a posteriori* estimate to the weights as $\mathbf{w}_{\text{MAP}}$, we can then use the Hessian matrix, $H$, at $\mathbf{w}_{\text{MAP}}$ to make a Laplace approximation to the posterior distribution: $p(\mathbf{w}|\mathcal{D}; \phi) = \mathcal{N}(\mathbf{w}_{\text{MAP}}, -H^{-1})$. We are now able to approximate the marginal likelihood, or evidence, of our model:

$$p(\mathbf{y}|\mathbf{x}, \phi) = \frac{p(\mathbf{y}|\mathbf{x}, \mathbf{w})\, p(\mathbf{w}|\phi)}{p(\mathbf{w}|\mathcal{D}; \phi)} \approx \frac{L \cdot \mathcal{N}(\mathbf{w}|\mathbf{u}, C)}{\mathcal{N}(\mathbf{w}|\mathbf{w}_{\text{MAP}}, -H^{-1})}. \tag{S1}$$

We then numerically maximize the model evidence with respect to the hyperparameters, $\phi$ (see Alg. 1 for pseudocode).

---

**Algorithm 1** Inference of weight trajectories and hyperparameters

---
**Require:** inputs $\mathbf{x}$, choices $\mathbf{y}$
**Require:** initial hyperparameters $\phi_0$, initial weights $\mathbf{w}_0$
 1: **repeat**
 2:      **repeat**
 3:          numerically optimize log-posterior w.r.t. $\mathbf{w}$, given current $\phi$ (Eq. 6)
 4:      **until** $\mathbf{w}$ converges     **return** $\mathbf{w}_{\text{MAP}}$ and the Hessian $H$
 5:      determine Laplace approximation to the posterior distribution, $\mathcal{N}(\mathbf{w}_{\text{MAP}}, -H^{-1})$
 6:      calculate model evidence (Eq. S1)
 7:      take one step in numerical optimization of log-evidence w.r.t $\phi$
 8: **until** $\phi$ converges     **return** current $\mathbf{w}_{\text{MAP}}$ and $\phi$

---

## C   Additional information about datasets studied (extension of Section 3)

Here we provide additional details about the datasets we analyzed in Section 3.

### C.1   Mouse visual detection task

We obtained the publicly available behavioral dataset associated with [4] from [6]. This dataset comprises decision-making data from 101 mice across seven laboratories We restrict our analysis here to the mice from one of the seven laboratories, resulting in the 13 animals mentioned in the text. This gave us a large enough cohort for examining commonalities in learning across animals, while also being small enough that we could fit all models to all animals with reasonable compute resources. For each of the 13 mice we studied, we fit all of our models to their first 6000 trials of training. During these trials, most animals progressed from chance level (50%) accuracy to above 70% (and often much higher) accuracy on the "easy" stimuli (different stimuli are introduced over time, so we plot the learning curves using only the "easy" stimuli that are present at all sessions); see Fig. S1 for the learning curves for these animals.

### C.2   Rat auditory discrimination task

We obtained the publicly available dataset associated with [2] from [1]. We analyzed the first 6000 trials from each of the 19 rats in this dataset (excluding pre-training trials, where the task rule was not enforced and all choices were rewarded; see [1]). During this stage of training, most rats progress from chance level (50%) accuracy to 55-70% (this is a particularly difficult task for the animals to learn due, to the delay between tones); see Fig. S2 for the learning curves for these animals.

# Learning curves: IBL animals

Figure S1: Accuracy on "easy" stimuli (different stimuli are introduced over time, so we plot the learning curves using only the "easy" stimuli that are present at all sessions) across 10,000 trials for 13 mice learning IBL visual detection task. Vertical lines indicate session boundaries; individual data points indicate accuracy for a given session.

# Learning curves: Akrami et al. rats

Figure S2: Accuracy across 6,000 trials for 19 rats learning auditory discrimination task. Vertical lines indicate session boundaries; individual data points indicate accuracy for a given session.

## D  Different learning rates for different weights (extension to Section 3.3)

In Fig. S3, we graphically explore the consequences of having different learning rates for different weights. Panel (a) corresponds to having a single learning rate for all weights. For a task like the IBL task, the stochastic gradient ascent update associated with the REINFORCE learning rule points in the direction of the gradient of the expected reward (since a REINFORCE update for any sampled stimulus-action pair will cause the animal to improve for any other stimulus). Thus, for a single learning rate parameter, the direction of the weight update is in the direction of the gradient of the expected reward landscape. In comparison, when different learning rates are permitted for each weight, the weight update direction may not be in the direction of steepest ascent in the reward landscape (panel (b)). Consequently, having separate learning rates for each weight allows for a broader range of learning trajectories in the expected reward landscape (panel (c)).

Figure S3: Schematic for the interpretation of having different learning rates for different weights. (a) Single learning rate for all weights. (b) Different learning rates for different weights. (c) In the case of different learning rates, the learning trajectory may not follow the path of steepest ascent in the reward landscape.

## E  Additional recovery analyses (extension to Section 3.1)

In Figure S4, we perform an equivalent analysis to that shown in Figure 2 and show that we can recover the REINFORCE with baseline learning rule parameters in simulation. Meanwhile, in Figure S5, we explore the effect of Model Mismatch. In panel (c), we show that AIC allows us to correctly identify the generative model from $RF_0$, $RF_1$, $RF_K$ and $RF_\beta$ when data is simulated from each of these models and then fit with each of these models.

## F  Noise and $RF_\beta$ model (extension to Section 3.4)

In Figure S6, we explore the necessity of including a noise term (i.e. allowing $\sigma_k \neq 0$) with the $RF_\beta$ model. As shown in Figure 3, the REINFORCE with baseline model does a good job of capturing the trial-to-trial updates in the weights governing an animal's policy. This then begs the question of whether it is necessary to include the noise term in the weight update of Equation 2. In Figure S6, we explore the effect of including and excluding a noise term when recovering weights generated from an $RF_\beta$ model with a small amount of noise (using the noise hyperparameters recovered from fitting real animal data), and we see that the model without noise fails to appropriately capture many of the trial-to-trial updates in the simulated weight trajectory. Additionally, AIC is worse for the noiseless model. Hence, it is important to include the noise term when fitting real animals' data with the $RF_\beta$ model.

Figure S4: Recovery of weights in t **(a)** Here we simulate a set of weights (in grey) using our REINFORCE with baseline model (similar to Fig. 2 for REINFORCE). We see that our recovered weights (plotted with a 95% credible interval) closely follow the true simulated weights. **(b)** Here we plot the learning components of the weights in (a). **(c)** Here we plot the noise components of the weights in (a). **(d)** Here we recover the six hyperparameters of the model. The $\sigma$ and $\alpha$ hyperparameters are recovered in $\log_2$ space. We found that our optimization has a more robust recovery of the quantity $\alpha \cdot \beta$ rather than the baseline hyperparameters $\beta$ directly. This $\alpha \cdot \beta$ quantity is recovered in non-log space. All error bars are calculated in non-log space and represent $\pm 1$SE. All hyperparameters are accurately recovered.

Figure S5: Exploring the impact of model mismatch. **(a)** Here we simulate four sets of weight trajectories (grey lines) from four of our models: no learning, REINFORCE with a single learning rate, REINFORCE with multiple learning rates, and REINFORCE with baselines. These four simulated trajectories (model indicated by row) are then recovered using each of the four models (model indicated by column; recovered weight trajectories are colored). Weights are simulated for 10,000 trials, and the depicted range of weight values is from -6 to +6. We can see that the weight trajectories recovered are all accurate, regardless of the simulated or recovery model. Gray background shading indicates that the recovery model matches the generative model. **(b)** The same as in (a) except only the learning component of the recovery is shown (the first column is the recovery under the no learning model, so there is no learning component to show). We see that despite the uniformity of the recovered weights, the recovered learning components vary greatly depending on the recovery model. **(c)** We calculate the AIC for each of the 16 models fit. Here each line corresponds to a row in (a), where the value plotted is recovered AIC minus the AIC of the recovery under the generative model. We see that for each line (each set of weights generated under a different generative model), the recovery under the matching model is the one with the lowest AIC. This suggests that our method can properly select between models.

Figure S6: Recovery of weights without noise. **(a)** Here we simulate weights from our REINFORCE with baseline model using very little noise (grey lines) and recover weights for a model without noise (colored lines). That is, we optimize for the $\alpha$ and $\beta$ hyperparameters while fixing the $\sigma$ to a negligible amount ($\sigma = 2^{-32}$ specifically). The model was simulated with a small amount of noise ($\sigma = 2^{-8}$), such that the learning contribution accounts for the vast majority of the weight trajectory. While S5 shows that models with noise all tend to recover roughly the same weight trajectories (independent of the learning model), fitting without noise can cause significant errors in the weight recovery (see red circle). **(b)** The same set of simulated weights as in (a), recovered with the standard REINFORCE with baseline model (with noise). We see that adding back noise allows the model to precisely capture the simulated weights. There is also a substantial decrease in model AIC ($\Delta$AIC > 100), indicating that noise clearly improves the model.

## G  Motivation for RAR and AAR Learning Rules (extension to Section 3.5)

In Fig. 4, we compared the REINFORCE learning rule to two closely related learning rules, which we termed "Reward-Agnostic REINFORCE" (RAR) and "Action-Averaged REINFORCE" (AAR). Here we provide additional motivation for each of these learning rules.

### G.1  Reward-Agnostic REINFORCE (RAR)

We obtain the Reward-Agnostic REINFORCE (RAR) learning rule by assuming that the animal is trying to minimize a cross-entropy loss, labeled $E_t$ at trial $t$, between the animal's choice probability, $p_{R,t}$ or $1 - p_{R,t}$, and the "correct" choice probability for the trial. This "correct" choice probability can be written as a Kronecker delta, $\delta_{\bar{y}_t,R}$, which is 1 if $\bar{y}_t = R$ and 0 otherwise. Similarly for $\delta_{\bar{y}_t,L}$.

$$E_t = -\delta_{\bar{y}_t,R} \log(p_{R,t}) - \delta_{\bar{y}_t,L} \log(1 - p_{R,t}) \tag{S2}$$

The gradient descent update for the cross-entropy loss is:

$$\Delta\mathbf{w} = -\alpha \frac{\partial E}{\partial \mathbf{w}} = \begin{cases} \alpha(1 - p_R)\mathbf{x} & \bar{y} = R \\ -\alpha p_R \mathbf{x} & \bar{y} = L \end{cases} \tag{S3}$$

Observe: this looks extremely similar to the REINFORCE update given in Eq. 4 and in Table S1 except that the RAR update does not depend on the action that the animal took, and instead all that matters is the side that was associated with the reward. For the same learning rate, we would expect that REINFORCE and RAR would give the same weight updates if the animal chose the correct action on every trial. Because of the differences in the updates for RAR and REINFORCE for error trials (for the reward function in $\{0, 1\}$, REINFORCE does not provide a weight update on error trials; in comparison, RAR gives an update regardless of the action that the animal took), an artificial agent using RAR should learn the task faster than agent learning using REINFORCE with the same learning rate.

### G.2  Action-Averaged REINFORCE (AAR):

By assuming that the animal maximizes the expected reward landscape directly instead of sampling its action from action space $\mathcal{A}$, we obtain the Action-Averaged REINFORCE (AAR) learning rule.

Specifically, let's consider the animal's expected reward on a given trial:

$$J(\mathbf{w}) = \sum_{\mathbf{s}} \mu(\mathbf{s}) \sum_{a} r(\mathbf{s}, a)\, \pi(a|\mathbf{s}, \mathbf{w}), \tag{S4}$$

where $\mu(\mathbf{s})$ is the animal's estimate of the probability distribution over stimuli, $\mathbf{s}$; $r(\mathbf{s}, a)$ is the reward associated with taking action $a$ when presented with stimulus $\mathbf{s}$; and $\pi(a|\mathbf{s}, \mathbf{w})$ is the animal's policy, which is parameterized by weights $\mathbf{w}$, and which gives the probability of taking action $a$ when presented with stimulus $\mathbf{s}$.

The gradient of the expected reward (where $\nabla = \nabla_{\mathbf{w}}$) is:

$$\nabla J(\mathbf{w}) = \sum_{\mathbf{s}} \mu(\mathbf{s}) \sum_{a} r(\mathbf{s}, a)\, \nabla \pi(a|\mathbf{s}, \mathbf{w}) \tag{S5}$$

If we assume that the animal uses only its most recent trial in order to estimate the state probability function, $\mu(\mathbf{s})$ is a delta function peaked at $\mathbf{s}_t$. Furthermore, let $\pi(a|\mathbf{s}, \mathbf{w})$ be the policy we considered in Eq. 1. Then we get the gradient ascent update:

$$\Delta\mathbf{w} = \alpha \nabla J(\mathbf{w}) = \alpha \sum_{a} r(\mathbf{s}_t, a)\, \nabla \pi(a|\mathbf{s}_t, \mathbf{w}) \tag{S6}$$

$$= \alpha\, [r(\mathbf{s}_t, R) - r(\mathbf{s}_t, L)]\, \pi(R|\mathbf{s}_t, \mathbf{w})\, \pi(L|\mathbf{s}_t, \mathbf{w})\, \mathbf{x}_t \tag{S7}$$

When $r(\mathbf{s}_t, R)$ and $r(\mathbf{s}_t, L)$ are in $\{0, 1\}$, and when we allow weight-specific learning rates, this reduces to the AAR update we provided in text (where, as in text, we have omitted the subscript $t$):

$$[\Delta\mathbf{w}]_k = \alpha_k \cdot p_{\bar{y}} \cdot \epsilon_{\bar{y}}(1 - p_{\bar{y}})[\mathbf{x}]_k \tag{S8}$$

Here, $\bar{y}$ is the correct answer for this trial and $p_{\bar{y}} \equiv p(\bar{y}|\mathbf{s}, \mathbf{w})$ is the probability that the animal makes the correct choice on this trial; $\epsilon_R = +1$; $\epsilon_L = -1$ (this is the same notation that we used in text). This rule was also proposed by Bak et al. [3], where it was called "RewardMax".

## H  Retrieved weights for mice & rats and $\mathrm{RF}_K$ & $\mathrm{RF}_\beta$ learning rules

In Figure Fig. S7, we show the weight trajectories (left column) and their decompositions into learning and noise components (right column) for 13 mice learning the International Brain Laboratory's visual detection task for the $\mathrm{RF}_K$ learning rule introduced in Eq. 4. We show the same for 13 rats learning the auditory discrimination task in Fig. S9 (omitting 6 of our 19 rats so as to fit the figure on a single page). In Fig. S7, the orange line corresponds to the weight the animal places on its bias, while the purple line shows its weight on the stimulus; in Fig. S9, the orange line indicates the weight the animal places on its bias, and the red and blue indicate the weights it places on each of the stimuli (the task is 2-dimensional). In each case, grey vertical lines partition sessions. In the right panel, the solid line shows the learning contribution of the $\mathrm{RF}_K$ learning rule to the trial-to-trial weight changes; the noise contribution to the trial-to-trial weight trajectories is indicated by the dashed line.

Crucially, we see that the phenomenon observed for the example mouse shown in Fig. 3 is observed across all animals in both cohorts: while the $\mathrm{RF}_K$ model can capture trial-to-trial changes in the weights that the mice and rats use to make their decisions (left panel), it is the noise component, *not* the learning component in this model that is responsible for capturing these changes (right panel; large shaded regions). This is particularly true for the bias weight for both sets of animals.

In Figures Fig. S8 and Fig. S10, we show the analogous plots to those mentioned above but when the $\mathrm{RF}_K$ learning rule is replaced with the $\mathrm{RF}_\beta$ learning rule of Eq. 5. We see that for all 13 mice in the IBL cohort, and for the majority of rats learning the auditory discrimination task, the $\mathrm{RF}_\beta$ rule is able to capture trial-to-trial weight changes with the learning component of this rule. Note: the initial contribution to the noise component (at $t = 0$) is removed from the right column in all plots for visibility, resulting in the learning components shown in the right panel to be vertically shifted relative to the full weight trajectories shown in the left panel.

IBL mice modeled with REINFORCE

inferred weight trajectories

decomposition into learning & noise

Figure S7: See figure description above.

IBL mice modeled with REINFORCE with baselines

inferred weight trajectories        decomposition into learning & noise

Figure S8: See figure description above.

rats modeled with REINFORCE

inferred weight trajectories

decomposition into learning & noise

tone A     bias

tone B

weights

0

-2

example rat

0    1000  2000  3000  4000  5000  6000
trials

Figure S9: See figure description above.

rats modeled with REINFORCE with baselines

inferred weight trajectories · decomposition into learning & noise

tone A · bias · tone B

weights

example rat

trials

Figure S10: See figure description above.