[Reviews · NeurIPS 2020]

Review 1

Summary and Contributions: This paper proposes a framework to infer the learning rule of animals, based on the REINFORCE policy gradient method. They validate their method via model recovery analysis, that they are able to recover the hyperparameters and weight trajectory with only behavioral data. They apply their method on two animal datasets and recover interesting traits of animal learning.

Strengths: Their study is novel to the best of my knowledge. I like this paper and I’m very excited about this topic. Few studies look at the training phase of animal learning or attempt to recover the policy-iteration process. They validate their model identifiability with simulation. I think it’s relevance to the NeurIPS community, as it applies method from reinforcement learning (REINFORCE) to neuroscience, and find interesting results of animal learning, which might inspire new RL algorithms.

Weaknesses: As it is pointed out by the author (line 148-151), the result strongly relies on the correct assumption of the learning model to be REINFORCE, which I think it’s a very strong assumption. It would be better supported by literature, showing animals can/are doing similar learning. Also as the authors pointed out, their model is descriptive. As the nature of a descriptive model, I feel like I don’t gain much insight from the model of how animals learn. For example, the authors found a non-zero update to the bias weight on incorrect trial, which explains the “incorrect” bahevior of repeatedly choosing the wrong option. This sounds like a “noise” in the behavior to me and the model also does not explain it further besides it being noise. Also, the author pointed out that the value function that the animals optimize for can be something other than expected reward, which I completely agree with. However, it seems their model does not provide any insight into their value function, besides saying it’s expected reward plus some noise. Please correct me if I misunderstand something, or I think it would be good if the authors can further concretely discuss how their model can provide more insights into animals for neuroscientists, and how it might be able to inspire machine learning algorithms. Also, the model recovery results only show one particular (hyper-)parameter setting, which can be recovered fairly well. I assume animals have lots of individual variabilities, and also variabilities across different tasks/datasets. So it is unclear if other parameter settings can also be well recovered, or it would be good to discuss if there are regions in the parameter space that can behave fairly similar. In addition, it would be good to show model identifiability results of RF_1, RF_K, RF_beta, i.e. if we simulate RF_1, will it be correctly identified as RF_1? It would be good to compare goodness-of-fit with some common value-iteration baselines that’s commonly used in animal neuroscience literature, e.g. RW and it’s variants (having two-learning rates, one for positive PE, one for negative; and/or adding a forgetting factor, that the value of the option(s) decaying towards zero/a parameterize value, with a decay parameter; and/or adding a side bias). I’m also interested to see if you simulate data with a fairly simple value-iteration method, and use your model to recover, what would you find? --- updates --- I appreciate authors effort into the response and being open to suggestions from the reviewers. With some of the new analyses, the paper will be stronger, however, it is hard to predict the results of those analyses, which could potentially make this paper weaker. I understand it is not reasonable to expect those new results within one-week. I'm keeping my original score, which is a borderline paper given the results currently presented.

Correctness: They are correct to the best of my knowledge.

Clarity: The paper is very well-written and I found it easy to follow. The figures are also very clear and easy to understand. One part that I found a little confused was 2.2 and 2.3. I was a little confused by the “placeholder” in equation 3. Would it be better if you introduce the learning models first? For 2.3, I wasn’t sure about the notation of the square brackets “[]”. I think the author can introduce REINFORCE even better, not assuming the readers all familiar with it, especially if you are also targeting neuroscientists as your audience.

Relation to Prior Work: The author discussed a few previous works attempt to look into animal policy-iteration and pointed out their contributions. Their unique contribution was clear to me.

Reproducibility: Yes

Additional Feedback: I’m a little confused about figure 5. Curve c also doesn’t look much alike curve a? I would also be interested to see how the model parameter might be correlated with, for example, the animal performance (accuracy). It would be interesting to see if some weight trajectory leads to better learning (accuracy, few-shot learning if there are switches in the task e.g. in Harlow task).


Review 2

Summary and Contributions: The paper proposes a really interesting method for inferring parameters in learning models and in particular the contribution of learning and noise terms. It provides an interesting way to analyse behaviour, even if the selected learning rules (mainly variations of REINFORCE) are not the most common in computational cognitive neuroscience - temporal difference reinforcement learning with softmax action selection or history-based models (as mentioned in the discussion) are more usual. It would be nice to see (at least in the discussion, if insufficient time to run simulations) how results could compare if using these models, which may help understanding if the negative bias result is some artifact specific to limitations of REINFORCE or some truly interesting phenomenon. I would suppose that stimulus uncertainty plays considerable role but especially for rodents, history-based patterns (such as win-stay, lose-shift) may be predominant and with appropriate parameters (e.g. low probability of shifting) may easily explain the negative bias result.

Strengths: The decomposition into learning and noise in evaluating learning models is particularly interesting. Although AIC is more commonly used, this sounds like a great and more specific alternative. Perhaps some discussion of comparison between the two could help. Furthermore, having the result replicated in two different datasets & species is clearly a plus. The contribution is clearly relevant for NeurIPS and at least some important aspects are novel, although broader importance of the results is a bit hard to evaluate due to limited range of learning rules explored.

Weaknesses: The main weakness is limitation of analysis to variations of REINFORCE learning rule to predict behaviour - a broader range, including some more commonly used rules should be explored. There is also no direct primary behavioural data presented to show the learning curves, which would be helpful to make sense of (model free) learning dynamics and hypothesize which models may be the most appropriate. Although I presume this could be found in references online, it would be helpful to add them at least to the SM.

Correctness: Yes, it seems correct. It's a bit strange that mean or median AIC values are not provided in Fig. 4c-d. Also, why is RF_beta missing in Fig. 4d? I think focusing on "example mouse" and presenting it in black is somewhat misleading and mean or median AIC values across mice should be shown instead (or in addition).

Clarity: Yes - it is explained reasonably well, although the pink trajectory in Fig. 5a is overlapping many times and a bit unclear - perhaps some continuous colour coding would help? It's also unusual that differences in learning rates between bias and stimulus (shown in Fig. 4b) are not discussed. Alphas seem much higher for bias and it would be good to discuss this. What exactly is R+B in Fig. 4c? (is it RF_beta?) Finally it's important not to confuse right (when it means direction) with right (meaning correct).

Relation to Prior Work: Generally yes; however, some clearly relevant work on (dynamic) parameter estimation of (dynamic) behavioural data is missing, such as https://papers.nips.cc/paper/2418-estimating-internal-variables-and-paramters-of-a-learning-agent-by-a-particle-filter.pdf and http://papers.neurips.cc/paper/3601-stress-noradrenaline-and-realistic-prediction-of-mouse-behaviour-using-reinforcement-learning.pdf As a result, the novelty assertion in lines 133-134 doesn't really hold.

Reproducibility: Yes

Additional Feedback: I thank the authors for their responses. I understand that REINFORCE is used loosely, but this still doesn't include a number or more common learning rules, such as TD learning. I simply think that both kinds of models should be included and compared openly (and making arguments that humans/animals use one and not the other is tricky and should generally be avoided, unless a large amount of evidence pointed that way, which I think is not the case...) Thanks for promising to address my other concerns. Look forward to reading the final version!


Review 3

Summary and Contributions: The authors develop a method to infer learning rules from observations of behavior. Specifically, they assume a specific parametric form for the relation between stimulus and behavior. They then test various learning rules which adjust these parameters (weights) over the course of training. Using a two stage inference procedure, the parameters of learning and weight trajectories are inferred from bahavior. This method is tested on simulated data and on two experimental tasks (in mice and rats). The authors show that a specific variant (multiple learning rates and baselines) provides a better fit to the experimental data compared to other learning rules tested.

Strengths: 1. Inferring learning rules is an important problem, and the authors leverage the recent availability of large behavioral datasets to address it. 2. The method is quite general with respect to the learning rules that can be tested. 3. The results show a “non-optimal” trajectory in weight space (Figure 5) which is an interesting observation.

Weaknesses: 1. Unless I missed it, there is no evaluation of the quality of the fit. Figure 4C,D provides the relative performance of the different rules, but there is no description of how well the best models fit validation sets. 2. The sub-optimal behavior finding calls for a more direct validation outside of the inference framework. 3. The simulations were only tested with matching generative and assumed learning rules. It would be useful to characterize the effect of a mismatch, as this is what is expected in real data.

Correctness: The results appear correct.

Clarity: The overall motivation and results are clearly presented. If the quality of fit is presented (Weakness #1), then it is not clearly presented.

Relation to Prior Work: Relevant prior work is mentioned.

Reproducibility: Yes

Additional Feedback: POST REBUTTAL UPDATE: I read all reviews and the author response. I was somewhat surprised that the rebuttal did not include any results. It is not realistic to expect a full revised paper within the one-week rebuttal period. But I think it is realistic to expect some results. For instance - correlating weights with accuracy shouldn't take much time. I think model mismatch is a crucial point. Even if there isn't time for a "broad exploration of hyperparameter space", there was time for a single example. All of these requests quickly add up - but I'm a bit concerned that the authors didn't do any of them during the rebuttal period. Furthermore - some of the results can qualitatively change my evaluation of the paper. If, for instance, the model fails completely once there is some mismatch (which is bound to exist in the data) - then it's hard to interpret the results of data analysis. Given this state - I'm keeping my original score. --------------------------------------------- 1. Line 112. Is D defined by a linear regression? 2. Simulated data: is it similar to animal behavior? Are there statistics to compare? 3. L153: How many trials/mouse? I think this is more relevant than the total. 4. Figure 4A: “outlier excluded”. How can there be an outlier for a quantity that is between [0,1]? 5. Figure 4D: Why is RF_beta not there? 6. Figure 5, trajectory a. Is this from RF_beta? The text (L168) describes this as the “retrieved weight trajectory for the animal”, but the retrieval is for specific assumptions. 7. L172: “trajectories for RF1 and RFk look very similar. How does this relate to Figure 5 b,c?


Review 4

Summary and Contributions: This paper proposes a method for modelling behavioural data in learning tasks. The paper is built upon the framework proposed in reference 19, and adds a learning element to the framework to enable deterministic changes in the learning weights. ================== post rebuttal ============================== The authors have mentioned that they will add the results regarding the first concern below to the final version, which would be great, thank you! For the second concern I still think the question of one vs two learning rates can be answered just by fitting the learning rules. Based on this I haven't changed the score.

Strengths: The paper is well-prepared, clearly written and the results are interesting. The mathematical framework is also novel to my knowledge, and can inspire model developments in other areas of computational modelling of decision-making.

Weaknesses: 1- Given the huge number of parameters in the model O(KT), it is important to see how the model performs compared to a baseline learning model *without* the noise. That would be, using ‘v’ in equation (1) instead of ‘w’ and comparing the obtained model in terms of AIC with the models presented in Figure 4c,d. From figure 3g it seems that the role of noise is negligible and there is not much benefit in using the full model beyond the learning rule (‘v’). 2- Whether the weights are updated using one or two learning rates could also be investigated by just fitting the learning rules (only using ‘v’ in equation 1) with different numbers of learning rates to the data and looking at their fit. Based on this, It is unclear to me what would be the insight about the animals’ behaviour that we have gained using this model that we couldn’t obtain from fitting just learning rules to the data.

Correctness: Yes

Clarity: Yes

Relation to Prior Work: Except for the addition of learning term (v), the computational framework is the same as the one in reference 19. This similarity, however, is NOT currently clear in the paper. I still think the contribution of the paper is sufficient, but it is essential that the authors be upfront about this similarity and make their contributions and ref 19 clear.

Reproducibility: Yes

Additional Feedback: In Figure 6d, it seems that the positive gap between the learning component of bias and the inferred trajectory systematically increases over time. Does it imply that the model lacks a time-dependent \beta parameter to absorb this variance? (similar to the logic presented in lines 175-178).

[Author Response · NeurIPS 2020]

We thank the reviewers for their detailed comments and helpful suggestions. We were delighted by the enthusiasm
expressed by all reviewers, and the unanimous decision to place our paper above the acceptance threshold. We are
grateful to Reviewer 2 for the confident & positive assessment of the paper, and humbly request that Reviewers 1, 3, &
consider raising their scores from 6/10 to 7/10 if they would like this work to reach the broader NeurIPS community.
We will first discuss general points raised by multiple reviewers, then address reviewer-specific comments.

• **Concerns that REINFORCE is the only learning rule considered** (**R1**, **R2**): We apologize for this confusion—
we were using the term "REINFORCE" loosely, but in fact our framework can easily be applied to the family of
policy-gradient learning rules. In fact, two of the rules we considered, AAR and RAR, are non-REINFORCE policy-
gradient rules. For example, the RAR rule is derived by optimizing a different objective function compared to that for
REINFORCE (see SM Eq.1). In the paper, we branded them as "variants" of REINFORCE, intending to make it easier
for readers unfamiliar with RL language. We will clarify this distinction in our revision.

• **Concerns that value function based models are not considered** (**R1**, **R2**): We agree that it would be exciting to
explore the space of other learning models, and one of our future directions is to replace the fixed baseline that we
currently use, with the value function (and, thus, incorporating a TD-component) in the $\text{RF}_\beta$ model. We also thank the
reviewers for suggesting a model comparison with variants of the Rescorla-Wagner model; this would be useful to
contextualize our results, and we would be happy to add this to the final paper. Meanwhile, we would like to point out
that while the computational cognitive science community tends to focus on TD-learning methods, a substantial number
of papers have proposed modeling decision-making behavior with variants of REINFORCE (e.g. Dayan & Daw (2008),
Kastner et al. (2019)); while Li & Daw (2011), provide support for the view that humans may use policy-gradient
methods instead of value prediction.

• **Concerns about model identifiability** (**R1**, **R3**): We agree that the identifiability of our models should be shown
explicitly. We will include this analysis in our revision, as well as an examination of the impact of model mismatch and
a broader exploration of hyperparameter space.

———————————————————————————————

**Reviewer 1** : (**a**) *The descriptive approach provides limited insight into how animals learn* — We apologize for
mischaracterizing our approach as purely descriptive; in fact, we view it as a platform for inferring the parameters
of normative models / testing normative hypotheses about animal learning. We would agree that our finding (that
negative baselines are required to account for animals' learning trajectories) is non-intuitive, and will add supplemental
analyses (e.g. conditioning on incorrect choice when bias is positive) to provide more insight into how the rule affects
choices. (**b**) *Correlating weights with empirical measurements like accuracy* — This is an excellent suggestion and is
straightforward to show; we will certainly include supplemental figures to address this.

**Reviewer 2** : (**a**) *Primary behavioral data to show the learning curves* — Great point, we will add learning curves
and other analyses to show that the inferred rules *do* indeed match behavior. (**b**) *Differences in learning rates for bias
and stimulus* — We should have thought of that! We will certainly discuss this in our revision, thank you for pointing
this out. (**c**) *Include additional references/lessen claims of novelty* — Thank you for the additional citations, we will
add these and remove the novelty claim in lines 133-4. (**d**) *Where is $\text{RF}_\beta$ in Figure 4c/d?* — Apologies, we mistakenly
labeled $\text{RF}_\beta$ as R+B, and $\text{RF}_K$ as R in Fig. 4c; we will fix this in the revision.

**Reviewer 3** : (**a**) *Evaluation of the quality of fit* — Thank you, we will add a quantification of performance in terms of
increase in log-likelihood on test set data. (**b**) *Sub-optimal behavior calls for direct validation* — Another great point,
we will use primary behavioral data to quantitatively validate the predictions made by the $\text{RF}_\beta$ model. (**c**) *Statistics of
simulated vs. animal data* — Again, great point, we will include this in our revision. (**d**) *Outlier excluded in Fig. 4a* —
Apologies, this phrase was a typo and will be removed. (**e**) *Why is $\text{RF}_\beta$ missing is Fig. 4d* — Apologies, we mistakenly
labeled $\text{RF}_\beta$ as R+B, and $\text{RF}_K$ as R in Fig. 4c; we will fix this in the revision.

**Reviewer 4** : (**a**) *Clarifying the relationship to other work (ref. 19)* — Thank you for pointing this out, we will absolutely
make the contributions of this work clear in relation to the computational framework from [19]. (**b**) *Comparing full
$\text{RF}_\beta$ model to $\text{RF}_\beta$ without noise* — This is an interesting point, and we will include a comparison of the current $\text{RF}_\beta$ fit
(REINFORCE with baseline, with noise) to one without a noise term in our revision. We respectfully disagree with the
comment that a noise term is unnecessary for the $\text{RF}_K$ models. In fact, it was by studying the structure of the noise
component that we were inspired to consider the $\text{RF}_\beta$ model. As was also observed, $\text{RF}_\beta$ is still not perfect for the
animal we display in the second dataset, but through examining the structure of the retrieved noise component, we
may be able to suggest a better normative learning rule (which may be exactly what was suggested by the reviewer – a
learning rule with time-dependent baseline parameters).

[Meta-Review · NeurIPS 2020]

I want to thank the authors for preparing the detailed rebuttal. This paper was discussed among all the reviewers during the post-rebuttal discussion phase. Overall, all the reviewers are excited about the research topic on inferring the learning rule of animals. There was a clear consensus that the paper should be accepted. The rebuttal did help clarify some of the reviewers' questions and steer their decisions towards acceptance. We hope that the authors will incorporate the additional results and changes, as mentioned in the rebuttal, when preparing the final version of the paper.